# Possible Role of Extracellular Vesicles in Hepatotoxicity of Acetaminophen

**DOI:** 10.3390/ijms23168870

**Published:** 2022-08-09

**Authors:** Martina Šrajer Gajdošik, Anamarija Kovač Peić, Marija Begić, Petra Grbčić, Kate E. Brilliant, Douglas C. Hixson, Djuro Josić

**Affiliations:** 1Department of Chemistry, Josip Juraj Strossmayer University, 31000 Osijek, Croatia; 2General Hospital Josip Benčević, 35000 Slavonski Brod, Croatia; 3Faculty of Medicine, University Juraj Dobrila of Pula, 52100 Pula, Croatia; 4Proteomics Core, COBRE CCRD, Rhode Island Hospital, Providence, RI 02903, USA; 5Warren Alpert Medical School, Brown University, Providence, RI 02903, USA

**Keywords:** liver, acetaminophen toxicity, non-parenchymal cells, extracellular vesicles, proteome

## Abstract

We examined proteomic profiles of rat liver extracellular vesicles (EVs) shed following treatment with a sub-toxic dose (500 mg/kg) of the pain reliever drug, acetaminophen (APAP). EVs representing the entire complement of hepatic cells were isolated after perfusion of the intact liver and analyzed with LC-MS/MS. The investigation was focused on revealing the function and cellular origin of identified EVs proteins shed by different parenchymal and non-parenchymal liver cells and their possible role in an early response of this organ to a toxic environment. Comparison of EV proteomic profiles from control and APAP-treated animals revealed significant differences. Alpha-1-macroglobulin and members of the cytochrome P450 superfamily were highly abundant proteins in EVs shed by the normal liver. In contrast, proteins like aminopeptidase N, metalloreductase STEAP4, different surface antigens like CD14 and CD45, and most members of the annexin family were detected only in EVs that were shed by livers of APAP-treated animals. In EVs from treated livers, there was almost a complete disappearance of members of the cytochrome P450 superfamily and a major decrease in other enzymes involved in the detoxification of xenobiotics. Additionally, there were proteins that predominated in non-parenchymal liver cells and in the extracellular matrix, like fibronectin, receptor-type tyrosine-protein phosphatase C, and endothelial type gp91. These differences indicate that even treatment with a sub-toxic concentration of APAP initiates dramatic perturbation in the function of this vital organ.

## 1. Introduction

Abuse of agents like alcohol and energy drinks, drug overdose, and different bacterial and fungal toxins are the most frequent causes of liver damage and failure of this vital organ [1]. Different drugs, such as analgesics, are also responsible for so-called drug-induced liver injury (DILI) [1,2,3]. Primarily for patients, but also healthcare providers and pharmaceutical companies, DILI represents a serious clinical and economical challenge [1,4,5,6].

Acetaminophen (APAP) is an analgesic and antipyretic drug. Therapeutic doses of APAP have only a few side effects [5]. However, overdoses of this medicament are one of the leading causes of DILI, especially in the presence of alcohol consumption [2,3,4,5,6,7].

Damage-specific markers found in plasma, serum, and urine, are frequently used for the diagnosis of liver failure. These would include enzymes like aminotransaminase, alanine aminotransaminase, gamma-glutamyl transferase, and 5′-nucleotidase [8].

Liver cells release extracellular vesicles (EVs) that can be detected in different body fluids, like plasma and urine. The composition of EVs shed by the liver depends on the state of this organ, suggesting that specific changes in EV components, which are mostly proteins and nucleic acids, could be used for diagnostic purposes [9,10,11,12]. Such an approach was suggested by the detection of some liver-derived proteins in exosome-like EVs purified from mouse and/or rat urine, and serum [9,13]. However, only a subpopulation of these EVs were liver-specific [14].

Proteomic analysis of EVs was most often carried out on primary hepatocyte isolates [9,12], identifying members of the liver-specific cytochrome P450 superfamily; members of uridine diphosphate-glucuranosyl-transferase (UGT) and glutathione S-transferase (GST) protein families; and a number of heat shock proteins [9,12,13]. Rodríguez-Suárez et al. [13] detected a higher concentration of the heat-shock proteins HSP90 and HSP70 in EVs released by primary hepatocytes, as well as in the serum of galactosamine-treated rats. These two proteins were identified as potential biomarker candidates for galactosamine-induced hepatitis [13].

Jaeschke and coworkers investigated APAP-induced mouse liver injury for almost twenty years. Their preference for mouse models stemming from the fact that mice are more similar to humans than other animal models [14,15,16,17]. Holman et al., gave preference to a rat model for their investigation of subtoxic alternations in early stages of acetaminophen-induced liver stress, occurring before hepatocellular injury [18].

We used a rat model to investigate changes in the proteomes of isolated liver EVs after treatment with a sub-toxic dose of APAP, prior to hepatocellular injury [14,18]. Changes to an EVs proteome after this kind of treatment indicate significant perturbation in the whole organ, even before other symptoms of toxic effects are observed. Already our preliminary experiments demonstrated striking differences in the size and distribution of EVs, as well as their protein composition [19]. In our present work, several proteins that are components of the extracellular matrix and non-parenchymal cells, such as Kupffer, stellate, and endothelial cells, were detected only in EVs of APAP-treated livers [19,20,21,22,23]. In contrast, some hepatocyte-derived proteins were not detected, or detected in relatively lower concentrations in EVs after APAP treatment, but high concentrations of these proteins were detected in EVs shed by normal rat livers or cultured hepatocytes [12,13,14,19]. These findings suggest that EVs collected after perfusion of APAP-treated livers are secreted not only by hepatocytes as parenchymal cells, but also by non-parenchymal liver components like stellate and endothelial cells, Kupffer cells, and non-resident macrophages.

In the present work, we isolated EVs by perfusion of the intact liver of experimental animals after treatment with a sub-toxic dose of APAP (500 mg/kg) in order to investigate EVs proteomes isolated from the entire complement of hepatic cells. Our focus was directed to the function and cellular origin of identified EVs proteins shed by different parenchymal and non-parenchymal liver cells and their possible role in an early response of this organ to toxic environments.

## 2. Results

### 2.1. Protein Identification and Comparison

Proteins detected by LC-MS/MS analysis of EVs shed by normal and acetaminophen liver are given in Appendix A as Appendix A. Proteins found only in EVs shed by the normal, untreated liver, are listed in Appendix A, and those detected only in EVs shed by APAP-treated liver are presented in Appendix A. In Appendix A, proteins detected in both samples are listed. Raw data are given in Appendix A in the Appendix A. Our preliminary data have been presented by Kovač-Peić et al. [19].

### 2.2. Protein Comparison According to Their Cellular Origin and Function

Further comparisons of proteins detected in EVs of a control liver and EVs shed after treatment of a liver with a sub-toxic dose of APAP are shown in Figure 1, Figure 2 and Figure 3. These Figures show a comparison of detected proteins according to their localization in cellular components (Figure 1), their molecular function (Figure 2), as well as their involvement in the different cellular processes (Figure 3). Furthermore, according to the localization in different liver tissues, proteins that are mainly localized in hepatocytes (as liver parenchymal cells), in Kupfer cells, and other liver non-parenchymal cells, including components of the extracellular matrix, are listed in Table 1 and Table 2.

As shown in Figure 1, proteins localized in the endoplasmic reticulum and ribosome, are mainly shed in EVs of liver cells before APAP treatment. A similar percent of proteins that are integral components of cellular membranes and cytoplasmic proteins were shed by the liver before, as well as after, APAP treatment. In comparison to EVs shed by the normal liver, in EVs shed after APAP treatment, a dramatic increase can be noted in the percent of plasma membrane proteins, followed by the increase in cell surface proteins, cytosolic proteins, and those localized in the cell–cell junctions.

Figure 2 shows a comparison of the proteins in EVs in normal and APAP-treated livers according to their molecular function. In EVs of a normal liver, a significant increase of proteins involved in oxidoreductase activity and enzyme binding can be noted. A similar percentage of proteins with hydrolase and transferase activity, and ATP binding, as well as metal ion binding, can be noted in both samples of the investigated EVs. The percentage of proteins involved in both nucleotide binding, especially GTP binding, and protein binding is significantly higher in APAP-treated vesicles.

The distribution of proteins according to their role in cellular processes in EVs shed by normal and APAP-treated livers is shown in Figure 3. In EVs shed by non-treated rat livers, the percentage of proteins involved in lipid metabolism, steroid metabolism, and xenobiotic metabolic process is significantly higher compared to the EVs shed by livers of APAP-treated rats. Proteins involved in liver development and translation are registered in both EVs samples in a similar percentage. The percentage of proteins involved in cell adhesion, immune system processes, and protein transport are considerably higher in EVs of APAP-treated animals. Interestingly, no proteins involved in the positive regulation of angiogenesis were detected in EVs shed by livers of non-treated animals. However, six percent of the proteins involved in this process were detected in EVs shed by livers of APAP-treated animals.

## 3. Discussion

The presented results show important changes in the composition of EVs shed by the liver after APAP treatment. These results had already been noticed after our preliminary experiments were recently published [19]. In order to achieve better interpretation of these results, and to elucidate the biological function of proteins shed by liver EVs before and after APAP treatment, additional bioinformatic analysis was performed.

### 3.1. Differences in Protein Patterns of EVs Shed by Normal and APAP-Treated Liver

The main proteins that are identified only in EVs of normal livers according to their location in liver cells are listed in Table 1. As shown in Figure 1, which presents a comparison of the unique proteins according to their cellular localization, EVs of the normal liver contain mainly proteins that are localized in the organelles, such as the endoplasmic reticulum, mitochondrion, and ribosomes. About the same percentage of cytoplasmic proteins, nuclear proteins, and those that are integral components of the membranes were detected in the EVs shed by livers of both control and APAP-treated rats. Non-treated livers shed EVs with a lower percentage of both cytoplasmic (31% in comparison to 52%) and cytosolic proteins (11% in comparison to 32%), but more significant are the relations between cell surface proteins (2% in comparison to 21%), the cell–cell junction (1% in comparison to 9%), and plasma membrane proteins (14% proteins in EVs shed by NL in comparison to 51% of proteins shed by livers of APAP-treated animals). Striking differences can be noted by further analyses, which are presented in Figure 2 and Figure 3. As shown in Figure 2, the analysis according to oxidoreductase activity shows a 25% drop in the amount of proteins that are shed by livers of APAP-treated animals (32% of proteins in EVs shed by NL in comparison to 7% of different proteins shed by livers of APAP-treated animals, a ratio of more than 4:1) and a difference in enzyme binding proteins (16:1). A similar result can be noted for the lipid metabolic process including steroid (12:1) and cholesterol metabolic process (7:1) (Figure 3). However, in the EVs shed by the livers of control animals, only a few proteins are detected that are involved in the following processes: cell adhesion (1:15), cell migration (2:7), inflammatory response (1:8), intracellular protein transport (1:9), positive regulation of angiogenesis (0:6), and transmembrane transport (1:11). The most significant general information is that most proteins detected in EVs of control livers are shed by hepatocytes (cf. Table 1 and Table 2). This picture is significantly changed after treatment with a sub-toxic dosage of APAP. Other proteins that are mainly localized in non-parenchymal liver cells or in extracellular matrix (especially fibronectin, cf. Table 1, Table 2 and Appendix A in the Appendix A) were detected. Further analysis of these differences leads to the conclusions presented below.

#### 3.1.1. Proteins That Are Detected Only in EVs Shed by Normal Liver

All proteins detected in EVs are either regularly shed into plasma by hepatocytes, or they are detected in parenchymal liver cells, mostly hepatocytes, before challenge with toxic substances [19].

Alpha-1-macroglobulin is known as an acute-phase plasma protein that is an inhibitor of different proteases [34]. Both protease inhibitors, alpha-1- and alpha-2- macroglobulins are frequently identified in different fractions extracted from rat liver plasma membrane preparations [55], but the significance of this finding is still not clear. However, this protein is, for the first time, identified as the main component of EVs that are secreted by the normal liver, while it is not detected in EVs that are shed after APAP treatment (see Table 1 and Table 2 and Reference [19]). The physiological relevance of this finding warrants further investigation.

As shown in Table 1, five members of the cytochrome P450 superfamily (2D26, 2D1, 2D4, 2C11, and 4A2) are shed by liver EVs of control animals. The concentration of these proteins in EVs shed by rat livers after APAP treatment is significantly lower, or they could not be detected et al. (see also Appendix A in the Appendix A). These numerous interrelated monooxygenases (over 50 at least) are expressed in the human liver, cf. Refs. [91,92,93], are mainly localized in hepatocytes, and they play a key role in the oxidation and metabolism of xenobiotics and endogenous compounds, and thus, are a primary defense against toxic reagents [91,92,93,94,95,96]. Additionally, this superfamily of enzymes has other important endogenous functions. It was demonstrated that the transfer of hepatocellular microRNA regulates cytochrome P450 2E1 production in rat kidneys (renal tubular cells). This process seems to be regulated by hepatic EVs [97].

We identified at least five members of the Glucuronosyltransferase superfamily (UGT) in EVs shed by non-treated livers that could not be detected in EVs after APAP treatment (cf. Appendix A and Appendix A in the Appendix A). These integral membrane glycoproteins are responsible for the process of glucuronidation, a major part of the so-called phase II metabolism. The reaction catalyzed by the UGT enzyme involves the addition of a glucuronic acid moiety to xenobiotics and is the most important pathway for the human body’s elimination of frequently prescribed drugs such as APAP. It is also the major pathway for the removal of foreign chemicals, such as most drugs, dietary substances, toxins, and endogenous substances [94,96].

The disappearance, or at least considerably lower concentration, of the members of these two superfamilies of enzymes (Cytochrome P450 and UDP-glucuronosyltransferase) that play a key role in the metabolism of xenobiotics in EVs that are shed after liver treatment with a sub-toxic concentration of APAP, is one of our key findings [19]. Bao at. al showed an age-dependent decrease in the expression of cytochrome P450 family members in mice after APAP treatment [98]. Recently, Rowland et al. [94] suggested the use of members of these two protein superfamilies as biomarkers for the absorption, distribution, metabolism, and excretion of xenobiotics in the body after their determination in human plasma after so-called liquid biopsy.

Retinol dehydrogenase 3 is localized in hepatocytes [24], and responsible for the maintenance of retinol homeostasis [88,89]. Again, a significant concentration of this enzyme was detected in EVs shed by livers of control animals but were not detected in EVs shed by livers of APAP-treated rats (see Table 1, Table 2 and Appendix A in the Appendix A).

In the liver, epoxide hydrolases catalyze the modification of arene—(both benzene and naphthalene) and aliphatic epoxides into less reactive and more water-soluble dihydrodiols. Although it seems to be also expressed in non-parenchymal cells [43], this important enzyme is not detected in EVs that are shed by rat livers after APAP treatment (cf. Table 1 and Table 2).

Cytochrome b5 reductase (CyB5R) is present in a relatively high concentration in the liver. The CyB5R catalyzes the reduction of ferricytochrome b5 to ferrocytochrome b5 using electrons from NADH [91]. Interestingly, only the membrane-bound form of CyB5R is, together with the membrane-bound form of cytochrome b5 (CyB5), involved in a variety of metabolic functions, including xenobiotic metabolism and detoxification, and in the bioactivation of carcinogenic drugs [30,32,91,92]. Both proteins are detected in EVs shed by normal rat livers, and they are not detected in EVs shed by livers of treated animals (cf. Table 1, Table 2 and Appendix A in the Appendix A).

#### 3.1.2. Proteins Identified in EVs Shed by Rat Liver after APAP Treatment

Significant changes in the protein pattern of EVs that are shed by the liver after APAP treatment can be observed. Altogether, 343 proteins are detected, and 117 of them are present only in EVs shed after APAP treatment (see Appendix A in Appendix A and Ref. [19]). Additionally, a relatively large number of shed proteins are of non-parenchymal origin (cf. Table 2). Figure 4 presents a possible coordinated reaction of the liver to this newly created situation, which seems to include the interaction between hepatocytes and non-parenchymal cells, thereby resulting in a different proteomic composition of newly shed EVs.

##### Fibronectin as a Component of Extracellular Matrix

Under pathological conditions such as liver injury, circulating levels of cellular fibronectin become disproportionally elevated. As early as the 1990s, Barkalow & Schwarzbauer [99] demonstrated that the interaction between fibronectin and glycosaminoglycans as components of extracellular matrices is essential for keeping tissue morphology and for cell adhesion. Following investigations confirmed the interactions between fibronectin and different cell populations, and noted that such interactions may be involved in the biological response to tissue injury. Aziz–Seible & Casey [35] investigated the effects of cellular fibronectin in the liver, specifically under pathological conditions (alcohol-induced injury). Under these conditions, cellular fibronectin was not synthesized by hepatocytes, but rather by endothelial cells and hepatic stellate cells, and the clearance of this matrix protein by other cells is impaired. It results in increased concentration of this glycoprotein throughout the liver, and in further interaction with hepatocytes and Kupffer cells. However, these authors did not investigate which specific organelles are involved in the secretion and transport of this protein. According to Reeves and Friedman, fibronectin synthetized in endothelial cells plays a key role in the activation of hepatic stellate cells in chronic liver injury by toxins, disorders of the immune system, and in viral and parasitic infections, as well as in some other rare liver diseases caused by genetic disorders such as galactosemia and Wilson’s disease [33].

Our results confirmed the preliminary finding that this glycoprotein was detected in very high concentrations in EVs that are shed by the liver after APAP treatment. Fibronectin was not detected at all in EVs shed by the non-treated liver (cf. Table 1 and Table 2). Already, a simple SDS-PAGE separation of proteins of these EVs followed by LC-MS/MS could demonstrate this fact (cf. Reference [19]). Purushothaman et al. [34] identified exosomal fibronectin, and they were able to demonstrate that this glycoprotein is a key ligand that binds glycosaminoglycan. This interaction is essential for further interaction between exosomes and target cells. However, the group did not perform further investigations in order to identify a specific fibronectin form (plasmatic or cellular) that could be mainly responsible for this interaction [20]. This kind of characterization of fibronectin, which is shed by liver EVs after APAP treatment, shall be the next main target of further investigation. However, the fact that high concentrations of this glycoprotein were detected in EVs shed by liver cells after APAP treatment supports the hypothesis that fibronectin plays an important role in starting disturbance of the liver after treatment with high doses of this analgesic [20,25].

##### Other Proteins Originated in Hepatocytes or Ubiquitously Localized in Liver Cell Populations

Na^+^/K^+^-transporting ATPase is an integral membrane protein of hepatocytes. It is essential for the transport of bile acids across the canalicular membranes of hepatocytes [36,37,100]. Together with fibronectin and major vault protein, this important transmembrane phosphoprotein is one of the most abundant proteins identified in hepatic EVs after APAP treatment (see Table 2), and its role in APAP toxicology shall be further investigated. 

There are some members of the solute carrier organic anion transporter (OATP) family identified in EVs shed only by APAP-treated livers (cf. Table 1 and Table 2), and of these, the OATP family member 1A4 is the most abundant. These drug uptake transporters are characterized by their broad substrate specificity. However, hepatotoxins influence their balance in the liver and the efflux of bile acids from hepatocytes into the blood [38]. The expression of these proteins has important implications in the disposition and delivery of targeted drugs [31], and their secretion after APAP treatment and following transport by EVs opens many interesting questions, like the influence of PTMs on the regulation of the function of these proteins [32].

The appearance of 5′-nucleotidase, a membrane anchored protein, can be interpreted as a evidence of liver injury. A 5′-nucleotidase test in serum is frequently used for the diagnosis of hepatobiliary and osseous diseases, and generally, for diseases connected with liver inflammation and malfunction [8,39].

Metalloreductase STEAP4 is localized in the plasma membrane but was also detected intracellularly in Golgi and endosomes of hepatocytes. Expression of this metaloreductase, which is involved in the transmembrane transport of iron and copper, is modulated by inflammatory cytokines, hormones, and other indicators of cellular stress, as well as in the response to inflammation and the metabolism of glucose and fatty acids [40,41].

Aminopeptidase N (also CD13) is localized in bile canaliculi and is essential for secretory activities, as well as for the development and functional maturation of the canalicular domain of hepatocytes [46]. Aminopeptidase N has also been discussed as a diagnostic marker for hepatocellular carcinoma [47]. Interestingly, this enzyme has also been known for decades as a receptor for coronavirus [42,44,45].

Major vault protein (MVP) plays an important role in the sorting of micro RNAs (miRNAs), which are, together with proteins, another important active component of EVs [48]. Micro RNAs are also detected in EVs shed by hepatocytes after APAP treatment [18], and the interaction between these two components, as well as their role during EVs-controlled cellular processes, seem to comprise one of the more interesting topics for future studies [49]. The multi-subunit-containing protein clathrin mediates the entrance of exosomes into cells by endocytosis and macropinocytosis. Together with caveolin- and lipid rafts-mediated endocytosis, clathrin-mediated endocytosis is one of the methods by which exosomal mRNA is transported, thereby enabling cell-to-cell communication. The high abundance of MVP and clathrin, proteins involved in the transport of EVs shed by APAP-treated rat livers, is a strong indication of an important disturbance in cellular function. Possible mobilization of some dormant cells can be initiated by the exchange of signal molecules like miRNAs and some proteins like MVP (cf. above) that are carried by these EVs [50,51]. 

Annexins build a family of phospholipid-binding proteins in a Ca^2+^-dependent manner. Annexins are frequently detected in EVs, and their secretion seems to be triggered by cellular stress caused by infection and inflammation [54]. It also seems that a part of vesicular annexins is only membrane-associated, and this binding may be mediated by Ca^2+^ ions. Microvesicular annexins also can bind other proteins and miRNAs that are transferred in exosomes and shed microvesicles [60]. Recently, annexin A1 was also identified as a specific marker for EVs that are shed directly from the plasma membrane [101]. This method for the secretion of annexins was also described in Ref. [54]. These low molecular weight annexins, namely annexin A1, A3, and A5, are found in significant concentrations in EVs shed by rat livers after APAP treatment. Annexin A6 was detected in both preparations of EVs, but the concentration of this high molecular member of the annexin family was significantly lower in EVs shed by livers after APAP treatment (see Ref. [23]).

##### Proteins of Kupffer Cells’ Origin and Proteins Originated from other Immune Cell Populations

Hepatocytes are the main cell population in the liver, and their proportion is about two-thirds of the total cell population. Furthermore, the liver is an important immunological organ, and its lymphocyte population is selectively enriched in natural killer (NK) and natural killer T cells. As resident macrophages, Kupffer cells play a crucial role in the first immune defense and modulation of liver injury. They are the first innate immune cells, and they protect the liver from infections. These macrophages are in close contact with passing lymphocytes, but they can make direct contact with hepatocytes and phagocytose in these apoptotic parenchymal cells [56,57]. According to Zigmond et al. [59], resident Kupffer cells in mouse livers challenged with toxic doses of APAP were significantly reduced, and they seem to have different ontogeny and functions in acute liver injury. Additionally, in the case of liver injury, liver-infiltrating macrophages can also develop into tissue-repairing cells [58]. As shown in Table 2, there are only a few proteins (like iron-carrying protein ferritin) that originate from EVs shed by immune cell populations of the non-treated liver. In contrast, some proteins that originated from these cell populations were detected in EVs shed by APAP-treated cells. (cf. Table 2). 

Band 3 anion transport protein is responsible for the blood’s capacity to transport carbon dioxide as soluble plasma bicarbonate [102]. At first, it is difficult to exclude the possibility that this protein has been detected in EVs as a contaminant coming from red blood cells in the liver. However, the perfusion protocol is specially designed to optimally remove this kind of contamination, and this protein was detected with a very high score (see Table 2). There are only a few studies of erythrophagocytosis by liver Kupffer cells, but this process seems to be promoted by both oxidative stress and inflammation [103]. 

Monocyte differentiation antigen, or CD14, is present in the serum of healthy animals, and in the culture supernatant of CD14 positive cells [104]. Soluble CD14 binds bacterial lipopolysaccharides [105] and they increase CD14 expression in Kupffer cells [61]. Monocyte differentiation antigen is a useful marker molecule for monocytes and macrophages [62]. The cell-surface receptor CD44 is a plasma membrane antigen that plays a role in cell–cell interactions, cell adhesion, and migration. In the liver, CD44 is mainly expressed in T and B lymphocytes, NKT cells, macrophages, and Kupffer cells. Both CD14 and CD44 play important roles in liver inflammations also caused by APAP overdose and the recovery of inflammation and fibrosis after acute liver injury [64]. Additionally, CD44 promotes both activation and migration of liver stellate cells after liver injury (see below and Ref. [66]).

##### Proteins of Stellate Cells

Although identified almost 150 years ago, the importance of hepatic stellate cells (HSCs), especially in liver injury and following fibrosis, and/or recovery, has been fully recognized only in the last twenty to twenty-five years [73,74]. These non-parenchymal cells are located in the subendothelial space of Disse, between hepatocytes and sinusoidal endothelial cells (see also Figure 4). This location enables their contact with other parenchymal and non-parenchymal cells such as hepatocytes, endothelial cells, and Kupffer cells, as well as direct interaction with the neighboring stellate cells. Quiescent HSCs are involved in normal liver development, vasoregulation, preservation of hepatocyte mass, and extracellular matrix homeostasis, as well as retinoid metabolism. Their role in extracellular matrix homeostasis, and especially in the primary cellular depot of retinoid (vitamin A), was the focus of early investigations [74]. The direct participation of quiescent HSCs in xenobiotic detoxification and oxidant stress response is possible, but it seems not to be their primary function in the healthy liver [68].

Following liver injury, HSCs become activated, and they trans-differentiate from (primarily) vitamin A-storing cells into proliferative, contractile, inflammatory, and chemotactic myofibroblasts [69]. Activated HSCs are characterized by enhanced production of extracellular matrix [74]. Consequently, the above discussed detection of fibronectin in EVs is one important indication that this matrix component is at least partially shed by activated HSCs. Extracellular signals from resident and inflammatory cells further modulate HSC activation, and this process seems to result in a dramatic change in the protein composition of shed EVs just a short time after treatment with a sub-toxic dose of APAP (cf. Table 1 and Table 2). The detection of some proteins that originate in stellate cells is a further indication of the increasing involvement of activated HSCs in the metabolic processes after APAP treatment. Similar results were published in a short report by Rani et al. [70], where the essential role of HSCs in APAP-induced liver injury was presented. Of particular significance is the fact that HSCs maintain a non-proliferative, quiescent phenotype only in normal livers, and that they become activated following liver injury. However, it should be taken into consideration, that their activation also takes place during in vitro culture [106], especially when the comparative proteomics of shed EVs of silent and activated stellate cells are investigated. Ji et al. [107] performed a comparative quantitative proteomic analysis of rat hepatic stellate cells by using iTRAQ labeling and 2D nano-LC-MS/MS. However, no pre-fractionation of isolated cells by isolation of organelles or separation of proteins by hydrophobicity was performed, and their results can be directly comparable only to a limited degree with those presented [30]. Published data also show additional and crucial involvement of miRNA and RNA-binding proteins in the above-discussed process of stellate cell activation [106,107,108].

4F2 cell-surface antigen heavy chain (LAT1 or CD98) biological function depends on different associated light chains, and include cell fusion, differentiation, proliferation, adhesion, and migration [109,110]. CD98 interacts with the transmembrane glycoprotein basigin (CD147), and both proteins induce integrin beta1 cell adhesion [21]. The same protein (basigin, CD147) also mediates the redistribution of CD98 that promotes cell spreading and tumorigenicity of hepatocellular carcinoma [111]. As an integrin–interactin protein, CD98 is also required for the growth of endothelial cells and for efficient angiogenesis [112]. The interaction of basigin with CD98 and its role in integrin beta1-mediated cell adhesion is discussed above. According to Zhang et al. [76], basigin also promotes the activation of hepatic stellate cells, and consequently, is also discussed as a potential therapeutic target in the treatment of liver fibrosis and cirrhosis.

The CD38 or ADP-ribosyl cyclase/cyclic ADP-ribose hydrolase 1 is an N-glycosylated membrane protein with one transmembrane domain. Localization of CD38 in hepatic stellate cells was also confirmed in both livers of patients with liver fibrosis and with chronic hepatitis [85]. Surprisingly, a recent hypothesis was raised that CD38, together with NADase, are also involved in the response to severe acute respiratory syndrome coronavirus 2 (SARS-CoV-2), which is responsible for the COVID-19 pandemic [113].

##### Proteins Originated from Hepatic Endothelial Cells

The liver’s response to inflammation, including sterile inflammation, results in an accumulation of leukocytes. In the liver, leukocytes bind to the endothelium, and the majority of their recruitment occurs through sinusoids. In response to any kind of liver injury, the recruitment increases as a consequence of endothelial activation [79]. Both intercellular adhesion molecules (ICAM-1) and integrins play a key role in this process [79,80]. In the liver, endothelial cells make up about 50 percent of all non-parenchymal liver cells [57]. One of the proteins that, in addition to HSC, can also be a component of EVs that are shed by endothelial cells, is CD98 [70].

Cell division control protein 42, or Cdc42, is a Ras superfamily GTPase is involved in many cellular activities, including cell–cell interaction and cancer progression [81]. Together with INF2, a formin protein that regulates actin polymerization and depolarization, Cdc42 regulates the dynamics of the integral membrane protein MAL2. This process is necessary for the apical targeting of liver endothelial cells [82,83].

Endothelial type Gp91-phox is an integral membrane protein that is detected in relatively high concentrations only in EVs shed by APAP-treated livers. It is one of two membrane components of a phagocyte NADPH oxidase. This enzyme complex is responsible for the production of superoxide anion, which generates other reactive oxygen species that are very toxic for the host cells. As a further consequence of this excessive production, inflammatory reactions and tissue injury occur [22,114]. Tetraspanin CD151 interacts with plasma membrane integrins and modulates their functions, e.g., in cell migration and endocytosis [115]. Tetraspanin is also an indicator of elevated T-cell activity and is an important marker of T-cell activation [116]. In the liver, CD151 is predominantly expressed in endothelial cells and supports lymphocyte adhesion mediated by the vascular cell adhesion protein 1 (VCAM-1 or CD106) in inflammation and hepatocellular carcinoma [86].

### 3.2. Possible Consequences of Observed Changes

The loss of important enzymes in EVs shed by APAP-treated rat livers, which are involved in the metabolism of xenobiotics, such as cytochromes P450, cytochrome b5, and cytochrome b5 reductase, is an indication of severe oxidative stress [103]. Furthermore, the loss of UGT enzymes that play a key role in the elimination of xenobiotics is also an additional sign of severe cellular stress [20,25,26,27,28,29].

The massive presence of fibronectin, MVP, Na^+^/K^+^-transporting ATP, and some other highly hydrophobic membrane proteins in EVs shed by APAP-treated cells is a further indication of a perturbation in this complex organ.

A high concentration of major vault proteins and their recently documented role in the sorting of miRNAs, another important active component of EVs, is the next highly intriguing finding [52]. Recently, Holman et al. [18] detected miRNAs in EVs shed by hepatocytes after APAP treatment.

Integrins and interacting proteins like talin and clathrin, as well as annexins, are frequently present during pathological changes in several organs. Especially in cancer, they are connected with poor prognosis [56,60,101,117,118] The disappearance of retinol dehydrogenase and the sudden appearance of proteins localized in stellate cells seems to be connected with their activation [106,107,108,109,110,119]. Furthermore, the appearance of some proteins localized in macrophages and Kupffer cells can also be interpreted as an indication of their increased activity [61,103,104,105,120].

## 4. Materials and Methods

### 4.1. Experimental Animals

Six-to-eight-week-old male Fischer-344 rats (three animals in each group, treated and non-treated; Jackson Laboratories, Farmington, CT, USA) were housed under standard conditions with access to food and water ad libitum. All animal experiments were performed following the guidelines of the National Institutes of Health and the Rhode Island Hospital Institutional Animal Care and Use Committee. All presented studies were approved by the Rhode Island Hospital Institutional Committee for Animal Care (Lifespan Animal Welfare Committee, Institutional Animal Care and Use Committee—CMTT# 0036-10, and Lifespan’s Animal Welfare Assurance #A3922-01, Lifespan, Rhode Island Hospitals’ and Health Services, Providence, RI, USA). Treated and control animals were euthanized 12 h after APAP oral administration [19,88,89,121].

### 4.2. Liver Perfusion

For perfusion, the portal vein of livers from untreated controls and animals treated with one dose of 500 mg/kg acetaminophen (Sigma Aldrich, Saint Louis, MO, USA) was canulated and the cannula attached to a peristaltic pump. The following perfusion at 37 °C with Hank’s balanced salt solution (HBSS) (Sigma Aldrich, Saint Louis, MO, USA) containing heparin to remove blood cells, the liver was removed and submerged in 80 mL of Hepatozyme serum-free medium (Thermo Fischer Scientific, Waltham, MA, USA). The isolated liver was perfused at 37 °C, according to the procedure described previously [19,88,89,121]. The experiments were performed in triplicate.

### 4.3. Isolation of EVs

EVs were harvested from a culture medium conditioned by perfusion through the isolated liver (see above) after centrifugation at low speed (1000× *g*, 5 min) (Beckman CS-6R, Beckman Coulter Inc., Indianapolis, IN, USA) and filtration through a 1.2 µm filter (Sartorius, Bohemia, NY, USA) to remove debris. As recently published, our previously used method for isolation of EVs from different rat organs and cell culture supernatant [19,89,121] was modified following the protocol of Hong et al. [89] by adding an additional two steps, namely sucrose gradient ultracentrifugation at 100,000× *g* and size-exclusion chromatography on Sepharose CL-2B [89,121]. In brief; after ultracentrifugation, an EVs-enriched pellet from each of three single livers was resuspended in 1 mL of PBS each and layered onto a 27%/68% sucrose cushion, then submitted to fractionation by size-exclusion chromatography (SEC) on an SEC column. The column was packed in our laboratory by use of a 10 mL plastic syringe packed with Sepharose CL-2B. Fractions of 0.5 mL each, corresponding to the dead volume of the column (between 3.5 and 8 mL), were concentrated by use of Amicon Ultra-15 membrane (100 Kda cutoff, Sigma Aldrich, Saint Louis, MO, USA). Finally, the protein content of isolated EVs and the size and form after the size-exclusion chromatography step were determined, and both the size and quality of the isolated EVs were checked by transmission electron microscopy (Morgagni 268, FeiTM, Hillsboro, OR, USA) [19,88,122].

### 4.4. Sample Preparation for Mass Spectrometry

In order to destroy EVs lipid envelopes, and to digest more proteins, especially very hydrophobic ones and ones with complex PTMs, several sample preparation methods before LC-MS/MS analysis were performed. In the case of the “gel-in-tube” proteolytic digestion [123], proteins together with proteolytic enzymes were incorporated into polyacrylamide gel. EVs fractions containing 100 µg of proteins were solubilized with 2% (*w*/*v*) SDS, 6 M urea, 25 mM NH_4_HCO_3_, pH 8.0, and further incubated at 37 °C for 30 min. The sample was then reduced at 56 °C for 1 h and alkylated with iodoacetamide at room temperature in the dark for 45 min.

The reduced and alkylated proteins were then incorporated into a polyacrylamide gel as described previously, and so-called “gel-in-tube digestion” was performed [23,117,123]. Proteolytic digestion was performed with trypsin (Sigma) in 40 mM NH_4_HCO_3_, 10% (*v*/*v*) acetonitrile overnight at 37 °C. Peptides were extracted from the gel using sequential extraction, as described previously [23,30]. The solutions were then combined and concentrated.

In some samples, in-gel deglycosylation was performed to facilitate the tryptic digestion of highly glycosylated proteins as previously described [30]. Gels were washed and sonicated, and then completely dried in a speed vacuum. Tryptic digestion was then performed by using the “gel-in-tube” method following the protocol described above (see also Refs. [19,117]). The experimental working scheme is provided in the Appendix A.

### 4.5. Protein Detection by LC-MS/MS

Tryptic peptides prepared using different sample preparation methods presented above were separated on a 75 µm × 12 cm column containing 3 µm Monitor C18 resin (Orochem Technologies, Inc., Lombard, IL, USA) with an integrated 10 µm ESI emitter tip (“Self-Pack” PicoFrit column; New Objective, Woburn, MA, USA) in the front of nano-LC-MS/MS. Solvent A was 0.1 M acetic acid in water and solvent B was 0.1 M acetic acid in ACN. Elution of separated peptides was performed using a linear gradient (0–70% solvent B over 60 min), operated at 200 nL/min on an Agilent 1200 HPLC (Agilent Technologies, Santa Clara, CA, USA). The nano-LC was hyphened in the front of an LTQ Velos Orbitrap Velos mass spectrometer (Thermo Scientific, San Jose, CA, USA) with a 1.8 kV ESI voltage. The nano-LC-MS/MS analysis was performed as previously described and the MS/MS spectra were searched against the Uniprot rat protein using the Mascot algorithm v.2.3.2 provided by Matrix Science. The exact procedures were previously given [23,114,121]. In brief, full MS scans were collected in the *m/z* range of 300–1700 at a nominal resolution of 60,000 followed with the acquisition of MS/MS spectra for the ten most abundant ions in the LTQ ion trap. Only ions having a charge state ≥ 2 were considered for collision-induced dissociation. Parameters for Mascot searches were as follows: trypsin enzyme specificity, 20 ppm mass tolerance, 2 possible missed cleavages. Search parameters specified a differential modification of oxidation on methionine and a static modification of carbamidomethylation (+57.0215 Da) on cysteine. Criteria for data filtering: Mowse score > 28 for all charge states, at least 2 peptides per protein, 1% peptide false discovery rate (FDR), and 1% protein FDR [23,114,121].

### 4.6. Gene Ontology Annotations of Detected Proteins

In order to identify differences in the protein profiles of isolated EVs, gene ontology (GO) analysis was performed using the QuickGO tool [124]. Uniprot Knowledgebase (UniProtKB) protein descriptions of unique proteins detected in EVs of normal and APAP-treated livers were imported into the software and proteins were annotated according to their biological process, molecular function, and cellular component, whereby a single protein can have several annotations.

Enriched GO terms were identified using Gene Ontology Enrichment Analysis and Visualization Tool GOrilla [125]. This tool provides GO enrichment analysis based on the minimum hypergeometric (mHG) statistical frame work. It enables comparison of two given protein sets, searching for GO terms that are enriched in the target set compared to the background set using the standard Hyper Geometric statistics. Lists of unique proteins detected in EVs from normal and treated liver were used alternately as a target and background set to obtain enriched GO terms for all tree ontologies (component, function and process). The threshold was set at the *p*-value of *p* < 0.001. The complete statistical analysis is given in Appendix A.

## 5. Conclusions

The effects of APAP toxicity on liver cells, especially on hepatocytes, are a formation of protein adducts, loss of glutathione, oxido-reductive cells, and necrosis. Both hepatocyte necrosis and recovery after the toxic influence of APAP are complex processes, decisively influenced also by other non-parenchymal cells like macrophages, Kuppfer cells, endothelial cells, and hepatic stellate cells. Our proteomic investigation pointed out that the toxic effect of APAP on rat livers was further delineated with the loss of enzymes involved in the metabolism of xenobiotics, which could be an indication of severe oxidative stress. Massive changes in the liver following APAP treatment are also marked with presence of fibronectin, MVP, Na^+^/K^+^-transporting ATP, and some other highly hydrophobic membrane proteins in EVs. These pathological changes were further corroborated by increased levels of the integrin family of proteins and their interacting proteins, such as talin and clathrin, and annexins. These findings are summarized in Figure 4.

Findings from this study suggest that the application of a subtoxic APAP dose of 500 mg/kg body weight can cause changes in the liver function delineated with specific changes on the proteome level. Further investigation of the proteins that are differentially expressed in EVs after treatment with APAP could reveal their significance and reversibility in liver injury and their future clinical potential.

## Figures and Tables

**Figure 1 ijms-23-08870-f001:**
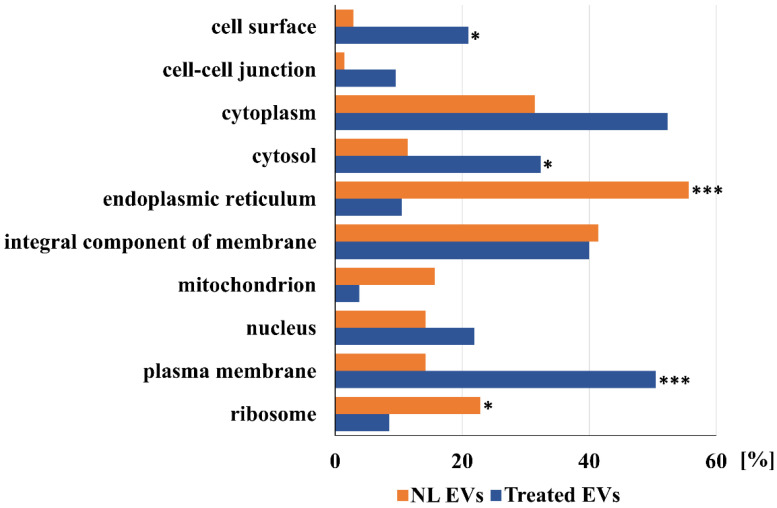
Comparison of unique proteins of normal (NL EVs) and acetaminophen treated (Treated EVs) liver EVs annotated according to their cellular component. For the enrichment of GO terms the threshold was set at the *p* < 0.001. Statistically significant enriched GO terms are marked according to the obtained *p*-value: * *p* < 0.001, *** *p* < 0.00001.

**Figure 2 ijms-23-08870-f002:**
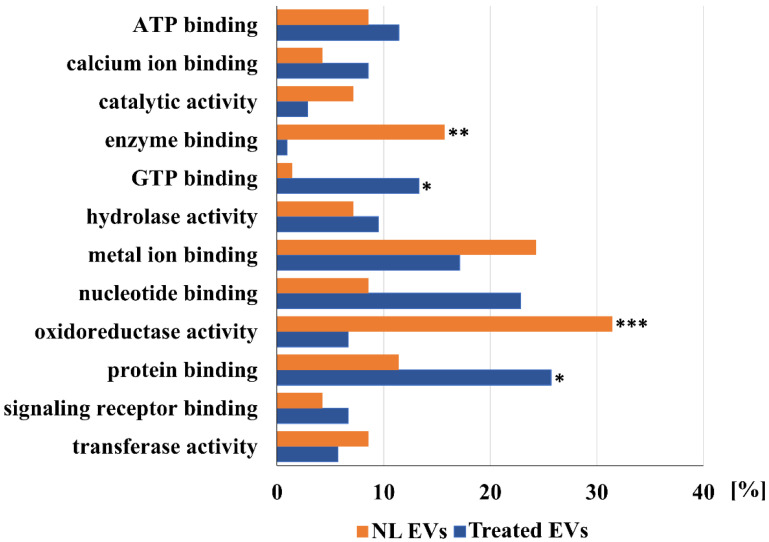
Comparison of unique proteins of normal (NL EVs) and acetaminophen treated (Treated EVs) liver EVs annotated according to their molecular function. For the enrichment of GO terms the threshold was set at the *p* < 0.001. Statistically significant enriched GO terms are marked according to the obtained *p*-value: * *p* < 0.001, ** *p* < 0.0001, *** *p* < 0.00001.

**Figure 3 ijms-23-08870-f003:**
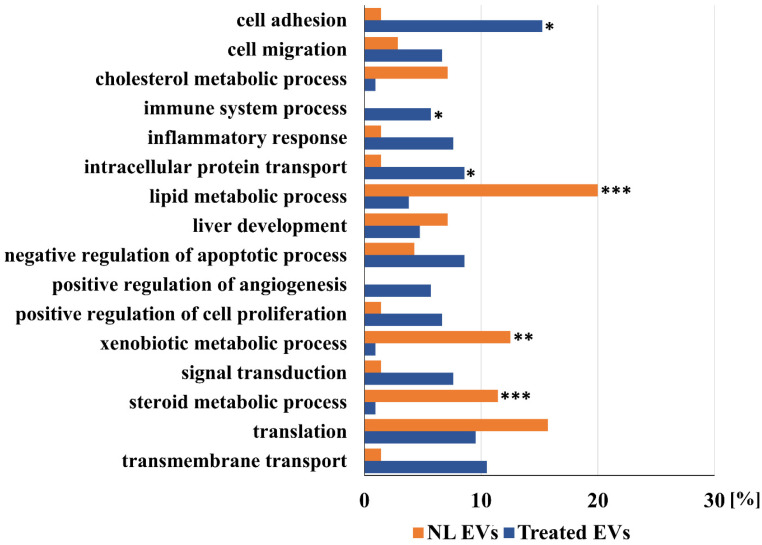
Comparison of unique proteins of normal (NL EVs) and acetaminophen treated (Treated EVs) liver EVs annotated according to their biological process. For the enrichment of GO terms the threshold was set at the *p* < 0.001. Statistically significant enriched GO terms are marked according to the obtained *p*-value: * *p* < 0.001, ** *p* < 0.0001, *** *p* < 0.00001.

**Figure 4 ijms-23-08870-f004:**
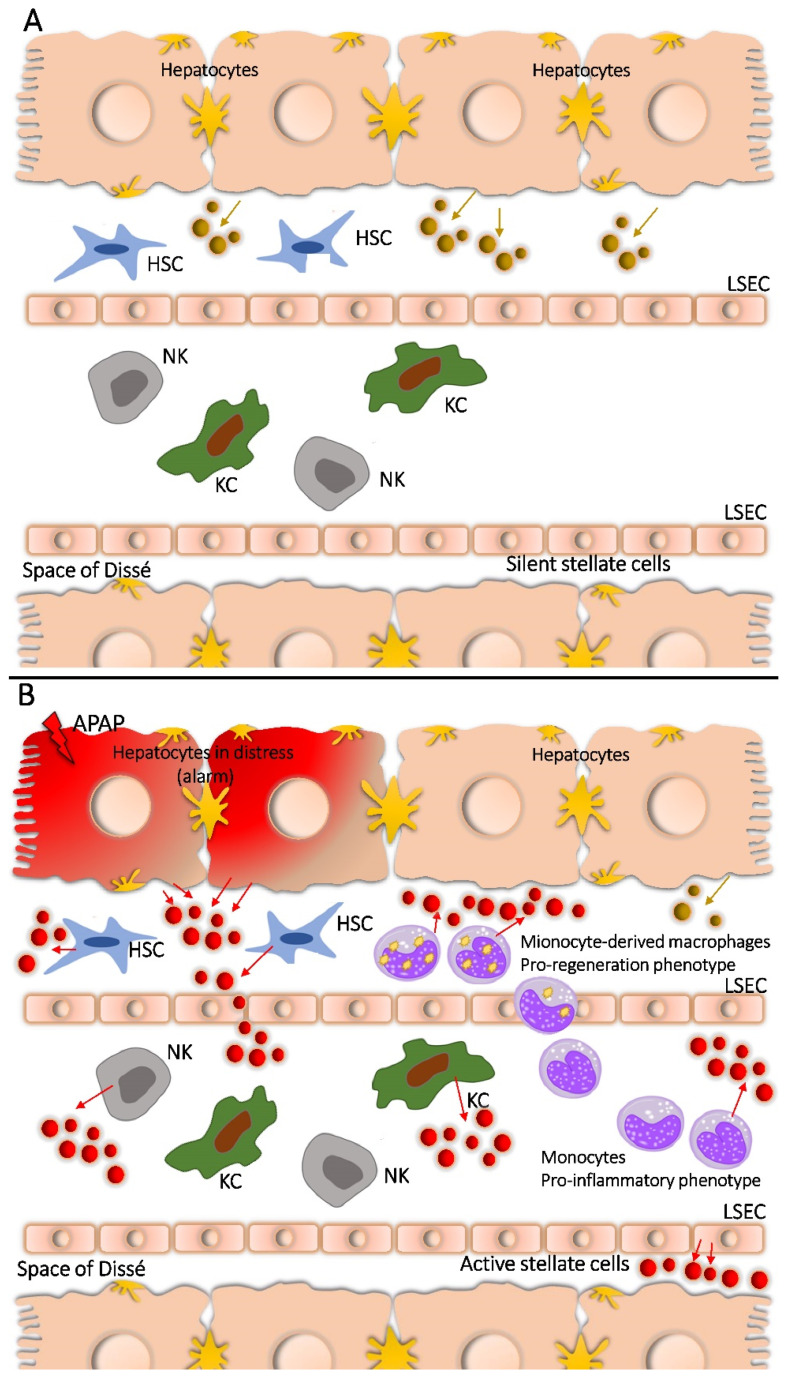
Liver and immune cells before (**A**), and after treatment with sub-toxic APAP dose (**B**). Distressed hepatocytes mobilize pro-inflammatory monocytes and other non-parenchymal liver cells like Kupffer cells, hepatic stellate cells, and endothelial cells. Hepatocyte necrosis, as a consequence of APAP overdose, did not occur under these experimental conditions, but changes in EVs proteomes shed by all involved liver cells indicated the liver was experiencing significant stress under the given conditions. KC—Kupffer cells, HSC—Hepatic stellate cells, NK—natural killer cells.

**Table 1 ijms-23-08870-t001:** List of proteins detected in EVs shed by normal (nontreated) liver according to their cellular origin.

UNIPROT	Protein Name	No. of Peptides Identified	% Coverage	Origin	Reference
Q63041	Alpha-1-Macroglobulin	25	22.53	Hepatocytes	[24]
P10634	Cytochrome P4502D26	18	44.4	Hepatocytes	[25,26,27,28]
P10633	Cytochrome P4502D1	14	32.54	Hepatocytes	[20,25,26,27]
Q64680	Cytochrome P450 2D4	8	14.73	Hepatocytes	[20,25,26,27]
P08683	Cytochrome P4502C11	6	18.2	Hepatocytes	[20,25,28,29]
P20816	Cytochrome P4504A2	4	8.9	Hepatocytes	[20,25,28,29]
P50170	Retinol dehydrogenase 3	8	29.97	Hepatocytes	[30,31,32]
F1M7N8	UDP-glucuronosyltransferase	7	16.97	Hepatocytes	[20,28,29]
P02793	Serum albumin	4	8.55	Hepatocytes	[23]
P27364	NADPH-dependent 3-keto-steroid reductase Hsd3b5	4	8.66	Hepatocytes	[33]
P20070	NADH-cytochrome b5 reductase 3	3	6.96	Hepatocytes	[25]
P00173	Cytochrome b5	3	6.23	Hepatocytes	[20,25,34]
07687	Epoxide hydrolase 1	6	20.88	Hepatocytes, Endothelial cells	[23]
P30839	Aldehyde dehydrogenase	6	17.77	Hepatocytes, Some non-parenchymal cells	[23]
P02793	Ferritin light chain 1	5	6.183	Hepatocytes Macrophages	[35]
P63259	Actin, cytoplasmatic 2	5	9.51	Ubiquitous	[23]

**Table 2 ijms-23-08870-t002:** List of proteins detected in EVs shed by APAP-treated liver according to their cellular origin.

Uniprot	Protein Name	No. of Peptides Identified	% Coverage	Origin	Reference
F1LST1	Fibronectin	16	10.81	Extracellular matrix	[36,37,38]
P06685	Na^+^/K^+^-transporting ATP-ASE	13	25.67	Hepatocytes & other polarized epithelial cells	[39,40,41]
P21588	5′-nucleotidase	6	13.54	Hepatocytes	[8,42]
O35913	Solute carrier org. anion transp. fam. member 1A4	5	6.092	Hepatocytes	[43,44,45]
Q4V8K1	Metaloreductase STEAP4	5	15.53	Hepatocytes	[46,47,48]
P15684	Aminopeptidase N (CD13)	4	4.974	Hepatocytes	[49,50,51]
Q62667	Major vault protein	13	25.67	Ubiquitous	[52]
P11442	Clathrin heavy chain 1	10	9.134	Ubiquitous	[53,54]
Q498D4	Talin	7	4.054	Ubiquitous	[55]
P07150	Annexin A1	6	18.18	Ubiquitous	[23,56,57,58,59]
P14669	Annexin A3	5	20.68	Ubiquitous	[23,56,57,58,59]
P14668	Annexin A5	2	8.464	Ubiquitous	[23,56,57,58,59]
B2RYB8	Integrin beta 2	5	15.84	Ubiquitous	[55,60]
P23562	Band 3 anion transport protein, (Solute carrier family 4 member 1 (SLC4A1))	5	8.588	Kupffer cells	[61,62]
P04157	Receptor-type tyrosine-protein phosphatase C (CD45)	3	22.92	Macrophages Kupffer cells	[20,63]
Q63691	Monocyte differentiation antigen (CD14)	2	4.17	Monocytes, Kupffer cells	[64,65,66,67]
D3Z257	Plexin B2	2	5.04	Monocytes	[68,69,70]
P26051	CD44	1	1.7	NKT cells (liver)	[71,72,73,74]
Q794F9	4F2 Cell surface antigen HC (LAT1 or CD98)	8	25.43	Hepatic stellate cells	[21,74,75,76,77,78]
Q64244	ADP-ribosyl cyclase/cyclic ADP-ribose hydrolase 1 (CD38)	3	8.581	Hepatic stellate cells	[79,80,81,82,83]
P26453	Basigin (CD147, HAb18G)	2	5.1	Hepatic stellate cells	[84,85]
Q8CFN2	Cell division control protein 42 (CD42)	2	3.22	Endothelial cells	[86,87]
Q9QZA6	CD151	1	5.929	Endothelial cells	[88,89,90]

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
