# Peer review of "Possible Role of Extracellular Vesicles in Hepatotoxicity of Acetaminophen"

_ijms, 2022, doi:10.3390/ijms23168870_

Round 1

Reviewer 1 Report

The manuscript has been modified, nonetheless there are some points to be better addressed as listed in the following.

The novelty and contribution of the proposed manuscript should be better assessed.

A detailed experimental working scheme of the experimental procedure and novelty of the approach used  could be useful.

Brand names of pharmaceuticals mentioning in the Introduction should be avoided unless necessary and justified. Line 57: “this application”, please rephrase for better clarity.

Please give date and number of the Ethical Committee authorization for studies on animals.

The statistical analysis is missing: please add thiese data to the manuscript.

Any shortening used should be exploited in full at first use, see e.g. “GO”, etc. Is it necessary thesection 2.6? Please give more details about the cited approach.

Too dated References should be avoided, e.g. the one dated 1976, etc.) in favor of more recet published papers.

Author Response

The novelty and contribution of the proposed manuscript should be better assessed.

Thank you. The novelty and contribution of this manuscript was pointed out in the conclusions of this manuscript and summarized in Figure 4.

A detailed experimental working scheme of the experimental procedure and novelty of the approach used could be useful.

We added the experimental scheme as a supplemental material. We also pointed out earlier that the detailed experimental approach was recently published in Reference 19.

Brand names of pharmaceuticals mentioning in the Introduction should be avoided unless necessary and justified. Line 57: “this application”, please rephrase for better clarity.

The brand names have been removed from the manuscript.

“This application” has been rephrased to “Such approach”

Please give date and number of the Ethical Committee authorization for studies on animals.

This information was already provided in response to one of the reviewers in the previous revision in Materials and Methods section as well as at the end of the manuscript.

All presented studies were approved by the Rhode Island Hospital Institutional Committee for Animal Care (Lifespan Animal Welfare Committee, Institutional Animal Care and Use Committee – CMTT# 0036-10, and Lifespan’s Animal Welfare Assurance #A3922-01, Lifespan, Rhode Island Hospitals’ and Health Services, Providence, RI, USA).

We can also send the copy of the document to the Editorial Office of the IJMS.

The statistical analysis is missing: please add thiese data to the manuscript.

Thank you, for your suggestion. Statistical analysis of enriched GO terms was performed using Gene Ontology Enrichment Analysis and Visualization Tool-Gorilla. The results of full analysis (6 tables with corresponding directed graphs) are provided in Supplementary Table 6, and Figures 1-3 were modified accordingly.

Any shortening used should be exploited in full at first use, see e.g. “GO”, etc. Is it necessary thesection 2.6? Please give more details about the cited approach.

Thank you for the suggestion, we overlooked providing the full name for the analysis. It is added in the title of the section 3.6. together with additional information on the use of this tool.

We do think that this section is necessary since it provides the information on the way the results were obtained.

Too dated References should be avoided, e.g. the one dated 1976, etc.) in favor of more recet published papers.

We usually do not refer to outdated papers. However, some references like Ref. 1-3. and Ref. 69.,70., and 76. are published early, but they are still of fundamental importance, giving information about liver physiology and the role of acetaminophen.

Reviewer 2 Report

Gajdošik et al. studied the effect of APAP on liver injury using bottom-up proteomics. For a research article, it is extremely lengthy and it is because there are too many information included that could have been avoided. I think discussing all the proteins, that were differentially expressed, that too in quite great details, in the discussion section is a bit overdo. I would recommend the authors to only include discussion for the ones that could be relevant to liver injury. Any treatment can impact the proteome of a biological system but might not carry biologically significant information. Additionally, the role of each protein is not required to be discussed unless they can be discussed in the context of APAP induced liver injury e.g., sentences like “Soluble CD14 binds bacterial lipopolysaccharides [113] and they increase CD14 expression in Kupffer cells [69]. Monocyte differentiation antigen is a useful marker molecule for monocytes and macrophages [70].” – I don’t see the point of including these information in the discussion section as it is not related to liver injury. Additional examples, how talin, clathrin, retinol dehydrogenase are relevant to APAP induced liver injury? Also, some comprehensive discussion can be made brief e.g., first section of 4.1.2.1.

1.      I don’t see the point of including a separate “aim of study section”. I do not understand what the previous reviewer 1 meant by his comments regarding this.

2.      Can the author explain why did they choose to do liver perfusion instead of an oral administration which will mimic the reality more closely than liver perfusion in a clinical setting?

3.      Page 3 line 120: can the author please mention the speed in g for low speed?

4.      Also, can the author clarify what buffer the pellets were in post filtration during ultracentrifugation and after that how they layered the 1 mL PBS fraction? Did they mean that they added the 1 mL PBS fraction to 27/68 (v/v) and then used it for SEC? I am interested to know these details.

5.      Does “tube gel” and “gel in tube” refer the same thing? If yes, please stick to one.

6.      Page 4 line 160: I don’t see any section with 2.5. Please check and correct this.

7.      It might be nice to have the full form of GO in 3.6

8.       Instead of comparing protein between treated and untreated groups based on location, function etc. it would be better if the author can do the same with the detected proteins that showed marked difference between treatment and no-treatment.

9.      Why certain cytochrome P450 proteins and UGT proteins (enzymes) were present in significant lower amount in APAP rat livers? Is APAP an inhibitor of these proteins?

10.  Tyrosine not thyrosine in page 15 line 517

Author Response

Gajdošik et al. studied the effect of APAP on liver injury using bottom-up proteomics. For a research article, it is extremely lengthy and it is because there are too many information included that could have been avoided. I think discussing all the proteins, that were differentially expressed, that too in quite great details, in the discussion section is a bit overdo. I would recommend the authors to only include discussion for the ones that could be relevant to liver injury. Any treatment can impact the proteome of a biological system but might not carry biologically significant information. Additionally, the role of each protein is not required to be discussed unless they can be discussed in the context of APAP induced liver injury e.g., sentences like “Soluble CD14 binds bacterial lipopolysaccharides [113] and they increase CD14 expression in Kupffer cells [69]. Monocyte differentiation antigen is a useful marker molecule for monocytes and macrophages [70].” – I don’t see the point of including these information in the discussion section as it is not related to liver injury. Additional examples, how talin, clathrin, retinol dehydrogenase are relevant to APAP induced liver injury? Also, some comprehensive discussion can be made brief e.g., first section of 4.1.2.1.

Thank you for the comment. We have additionally excluded some protein information from the discussion as suggested.

However, some information are still necessary in order to document the cellular localization of key proteins in parenchymal and non-parenchymal cells, the number of transmembrane and intermembrane domains and their PTS. Proteins that are secreted by non-parenchymal cells suggest the activation of these cells after APAP treatment of animals.

  1. I don’t see the point of including a separate “aim of study section”. I do not understand what the previous reviewer 1 meant by his comments regarding this.

                  As the reviewer states, this was done to respond to the request of the other reviewer.

  1. Can the author explain why did they choose to do liver perfusion instead of an oral administration which will mimic the reality more closely than liver perfusion in a clinical setting?

Thank you for the comment. We now see that it was not specifically stated that the APAP administration was per oral. This information was added to the manuscript. But liver perfusion was performed to obtain EVs after oral administration of APAP, i.e. after induced liver injury.

  1. Page 3 line 120: can the author please mention the speed in g for low speed?

                  The information has been added to the text.

…after centrifugation at low speed (1000 x g, 5 min) and filtration through

  1. Also, can the author clarify what buffer the pellets were in post filtration during ultracentrifugation and after that how they layered the 1 mL PBS fraction? Did they mean that they added the 1 mL PBS fraction to 27/68 (v/v) and then used it for SEC? I am interested to know these details.

As stressed in the text and in Reference [19] we just followed already published protocols for isolation of extracellular vesicles (EVs). Their isolation and difference in form und size between is also documented in the Figure 1 of the mentioned paper. 

  1. Does “tube gel” and “gel in tube” refer the same thing? If yes, please stick to one.

Thank you for the suggestion. You are right and we made corrections troughout the text using the term “gel-in-tube”

  1. Page 4 line 160: I don’t see any section with 2.5. Please check and correct this.

We apologize, it was overlooked due to several reorganizations of the manuscript. It is now corrected.

  1. It might be nice to have the full form of GO in 3.6

Thank you for the suggestion, we overlooked providing the full name for the analysis and apologize for that. It has been now added to the text.

  1. Instead of comparing protein between treated and untreated groups based on location, function etc. it would be better if the author can do the same with the detected proteins that showed marked difference between treatment and no-treatment.

The aim of this study was to demonstrate that APAP treatment initiates activation of non-parenchymal cells. The activated cells shed EVs that contain specific proteins that were not shed by hepatocytes. At the other hand, APAP seems to inhibit shedding of specific proteins that are localized in hepatocytes. Cf. also our comment above.

  1. Why certain cytochrome P450 proteins and UGT proteins (enzymes) were present in significant lower amount in APAP rat livers? Is APAP an inhibitor of these proteins?

Thank you for this comment. We added additional reference to the manuscript that clearly documents influence of APAP on the expression of these proteins in mouse liver. Additionally, these proteins were not detected in EVs shed by livers of treated animals.

Bao, Y.; Wang, P.; Shao, X.; Zhu, J.; Xiao, J.; Shi, J.; Zhang, L.; Zhu, H.-J.; Ma, X.; Manautou, J.E.; et al. Acetaminophen-Induced Liver Injury Alters Expression and Activities of Cytochrome P450 Enzymes in an Age-Dependent Manner in Mouse Liver. Drug Metab. Dispos. 2020, 48, 326–336, doi:10.1124/dmd.119.089557.

  1. Tyrosine not thyrosine in page 15 line 517

                  Thank you, it has been corrected.

Round 2

Reviewer 2 Report

Thank you for addressing my comments. Congratulations!